# Mechanistic study of plastic monomers in gestational diabetes mellitus: A network toxicology and molecular docking approach

**Yingying Feng, Tingting Huang** *

Department of Obstetrics, Maternal and Child Health Hospital of Hubei Province, Tongji Medical College, Huazhong University of Science and Technology, Hongshan District, Wuhan, China

* tingtinghuang312@163.com

## Abstract

Plastics are widely used in various fields such as food packaging, textile fibers, building materials, and transportation. Although the relationship between plastic additives and diseases has been reported, there is limited research on the association between plastic monomers (PM) and gestational diabetes mellitus (GDM). This study aims to investigate the link between environmental PM and GDM. By employing advanced network toxicology and molecular docking techniques, we successfully elucidated the molecular mechanisms by which PM may induce GDM. Utilizing databases such as PubChem, SEA, Super-PRED, SwissTargetPrediction, PharmMapper, Gene Cards, and OMIM, we identified potential targets associated with the disease. Further analysis using STRING and Cytoscape software helped determine the core targets most significantly related to these metabolic disorders. Additionally, Gene Ontology and Kyoto Encyclopedia of Genes and Genomes pathway enrichment analyses were conducted using the David database to characterize these core targets. Finally, molecular docking with CB-Dock2 was used to validate the binding affinity of PM to these target proteins. Our findings suggest that PM may potentially induce GDM by modulating the insulin signaling pathway through STAT3, AKT1, and TP53. In summary, this work provides novel insights into the mechanisms by which environmental pollutants may trigger GDM, thereby laying a theoretical foundation for disease prevention and treatment. It offers valuable references for the safety evaluation of plastics, urging food safety regulatory agencies to strengthen oversight and encouraging the public to reduce plastic usage.

## 1. Introduction

The widespread use of plastics has led to human exposure to their associated toxic chemicals [1]. Structurally, plastics constitute complex mixtures of monomers, additives (e.g., plasticizers, colorants), and non-intentionally added substances

**Data availability statement:** All relevant data are within the manuscript and its Supporting Information files(Supplementary_Material.xlsx). All raw data required to replicate the results of this study are provided in the supplementary file 'Supplementary_Material.xlsx'. All data analyzed in this study were obtained from publicly available sources. Complete descriptions of data collection, processing, and analytical methods are provided in the Methods section.

## 2. Methods

### 2.1 Identification of PM targets

The SMILES and SDF structural information of PM was retrieved from the PubChem database (https://pubchem.ncbi.nlm.nih.gov/). Potential human (Homo sapiens) targets of PM were systematically screened using the SEA (https://sea.bkslab.org/) (23), Super-PRED (https://prediction.charite.de/) (24), SwissTargetPrediction (http://www.swisstargetprediction.ch/) (25) and PharmMapper (https://lilab-ecust.cn/pharmmapper/index.html) (26) databases. Target names were standardized via the UniProt database, and duplicate entries across the four databases were removed. Detailed information on these PM, including their molecular formulas, molecular weights, and SMILES structures, is provided in Table 1.

Table 1. Detailed information of PM. Molecular Name CAS ID Formulas SMILES structures Acrylonitrile 107-13-1 C3H3N C=CC#N Adipic acid 124-04-9 C6H10O4 C(CCC(=O)O)CC(=O)O Bisphenol A 80-05-7 C15H16O2 CC(C)(C1=CC=C(C=C1)O)C2=CC=C(C=C2)O Butadiene 106-99-0 C4H6 C=CC=C Caprolactam 105-60-2 C6H11NO C1CCC(=O)NCC1 Diphenyl Carbonate 102-09-0 C13H10O3 C1=CC=C(C=C1)OC(=O)OC2=CC=CC=C2 Ethylene glyco 107-21-1 C2H6O2 C(CO)O Hexamethylenediamine 124-09-4 C6H16N2 C(CCCN)CCN Methyl Methacrylate 80-62-6 C5H8O2 CC(=C)C(=O)OC Styrene 100-42-5 C8H8 C=CC1=CC=CC=C1 Terephthalic acid 100-21-0 C8H6O4 C1=CC(=CC=C1C(=O)O)C(=O)O

### 2.2 Acquisition of GDM related targets

GDM associated targets were extracted from the GeneCards (https://www.genecards.org/) and OMIM (https://www.omim.org/) databases using the keyword GDM, restricted to Homo sapiens. We selected GDM targets with a relevance score ≥ 10 from the GeneCards database. Duplicates between the two databases were merged and deduplicated. The intersection of PM targets and GDM-related targets was identified as potential key

(e.g., impurities in raw materials and by-products of polymer production) [2,3]. A significant concern is that plastics may incorporate over 16,000 monomers and additives, with toxicological hazard data severely lacking for the majority of these chemicals [4,5]. Of particular concern is the fact that most of these chemicals are not covalently bonded to the polymer matrix. This allows them to be released into environmental compartments through migration, volatilization, or leaching processes at various stages of the plastic lifecycle. Consequently, these substances can transfer into packaged contents (e.g., food), human exposure microenvironments (e.g., indoor air and dust), and natural environments (e.g., water bodies and soil), thereby establishing plastic as a significant source of human exposure to synthetic chemicals [6,7]. A prominent example is bisphenol A, a monomer used in plastics. It has been widely detected in human urine, exhibiting a prevalence exceeding 92% in the general U.S. population [8]. This high detection rate confirms the ubiquity of exposure. This exposure facilitates the bioaccumulation of these chemicals and is associated with diverse health risks, including carcinogenicity, mutagenicity, reproductive toxicity, specific target organ toxicity, endocrine-disrupting effects, and ecotoxicity [9–12]. Such hazards are corroborated by occupational and community-based epidemiological studies: plastic production workers exhibit significantly higher incidences of cancer and pulmonary diseases compared to the general population; furthermore, residents near plastic manufacturing facilities demonstrate elevated rates of preterm birth, low birth weight, childhood leukemia, asthma, chronic obstructive pulmonary disease, cardiovascular disease, traffic injuries, and mental health issues [1]. Notably, certain additives, such as phthalates, chlorinated paraffins, and brominated flame retardants, have been identified as carcinogens [13,14]. However, research focus remains unbalanced, predominantly centered on plastic additives, with comparatively limited investigation into the health risks posed by the monomers themselves.

Plastic monomers (PM) such as acrylonitrile, adipic acid, bisphenol A, butadiene, caprolactam, diphenyl carbonate, ethylene glycol, hexamethylenediamine, methyl methacrylate, styrene, and terephthalic acid serve as fundamental building blocks for common plastics like ABS, PET, PC, PA66, PA6, and PMMA. The extensive use of these plastics across industries including food packaging, textiles, construction materials, and transportation leads to inevitable environmental release of monomers, resulting in widespread human exposure [15]. Toxicological assessments indicate that most PM possess potential toxicological hazards. For instance, common monomers like bisphenol A and styrene are not only confirmed reproductive toxicants but are also classified as known or suspected carcinogens; likewise, phthalates are recognized reproductive toxicants, and vinyl chloride is a confirmed carcinogen [16]. Despite this recognition, substantial knowledge gaps persist regarding PM, particularly concerning their health impact mechanisms and risk levels for vulnerable populations.

Pregnant women represent a uniquely susceptible population due to their distinct physiological state, exhibiting heightened sensitivity to chemical exposures that can induce multiple adverse effects on maternal health and pregnancy outcomes [17,18]. Gestational diabetes mellitus (GDM) is the most common pregnancy complication

mediators of PM toxicity in GDM pathogenesis. 2.3 Construction of protein-protein interaction networks Common targets shared between PM and GDM were analyzed using the STRING database (https://cn.string-db.org/) to construct a protein-protein interaction (PPI) network with a high-confidence interaction threshold (confidence score ≥0.9) (27). The network was visualized and analyzed using Cytoscape_ v3.10.3 software, with key hub genes identified via the Maximal Clique Centrality, Maximum Neighborhood Component, Degree, Closeness, Betweenness and Stress algorithm.

**Funding:** The author(s) received no specific funding for this work.

**Competing interests:** The authors have declared that no competing interests exist.

among women worldwide, with its incidence demonstrating a consistently upward trend [19,20]. Beyond traditional risk factors such as genetic predisposition, dietary patterns, and physical activity, environmental chemical pollutants are increasingly recognized as emerging drivers of metabolic disorders like obesity and diabetes, posing particular risks to susceptible groups including pregnant women and children [21,22]. Research suggests that environmental pollutants may contribute to GDM pathogenesis through various biological mechanisms, including the induction of insulin resistance, impairment of β-cell function, activation of inflammatory pathways, and the promotion of oxidative stress [19]. Notably, bisphenol A and its substitutes have been implicated in GDM risk. For example, a comparative study found a significant association between elevated urinary bisphenol A levels and increased GDM risk [23]; furthermore, recent prospective cohort studies in China indicate that bisphenol S and bisphenol AF may also be potential risk factors for GDM [24]. This evidence suggests that other widely used, structurally diverse PM with potential metabolic-disrupting activity may also play a role in GDM etiology. However, epidemiological and experimental studies directly investigating the association between PM exposure and GDM remain scarce. The causal relationship and specific molecular mechanisms underlying this potential link are unclear and warrant urgent investigation.

To address this critical research gap and preliminarily assess the potential role of PM in GDM pathogenesis, this study employs an integrated strategy combining network toxicology and molecular docking. Our aim is to systematically elucidate the potential pathophysiological links between PM exposure and GDM development. Specifically, we utilize network toxicology to construct interaction networks between PM and metabolic regulation, and employ molecular docking to simulate the binding characteristics and affinities between PM and key molecular targets (e.g., receptors, enzymes). This integrated approach facilitates the generation of mechanistic insights at systemic and molecular levels, clarifying how PM might disrupt metabolic homeostasis by perturbing specific signaling pathways. It thereby provides a novel computational toxicology paradigm for assessing the endocrine and metabolic disruption potential of complex environmental chemicals.

## 2. Methods

The flowchart for this study is illustrated in Fig 1.

### 2.1 Identification of PM targets

The SMILES and SDF structural information of PM was retrieved from the PubChem database (https://pubchem.ncbi.nlm.nih.gov/). Potential human (Homo sapiens) targets of PM were systematically screened using the SEA (https://sea.bkslab.org/) [25], Super-PRED (https://prediction.charite.de/) [26], SwissTargetPrediction (http://www.swisstargetprediction.ch/) [27] and PharmMapper (https://lilab-ecust.cn/pharmmapper/index.html) [28] databases. Target names were standardized via the UniProt database, and duplicate entries across the four databases were removed. Detailed information on these PM, including their molecular formulas, molecular weights, and SMILES structures, is provided in Table 1.

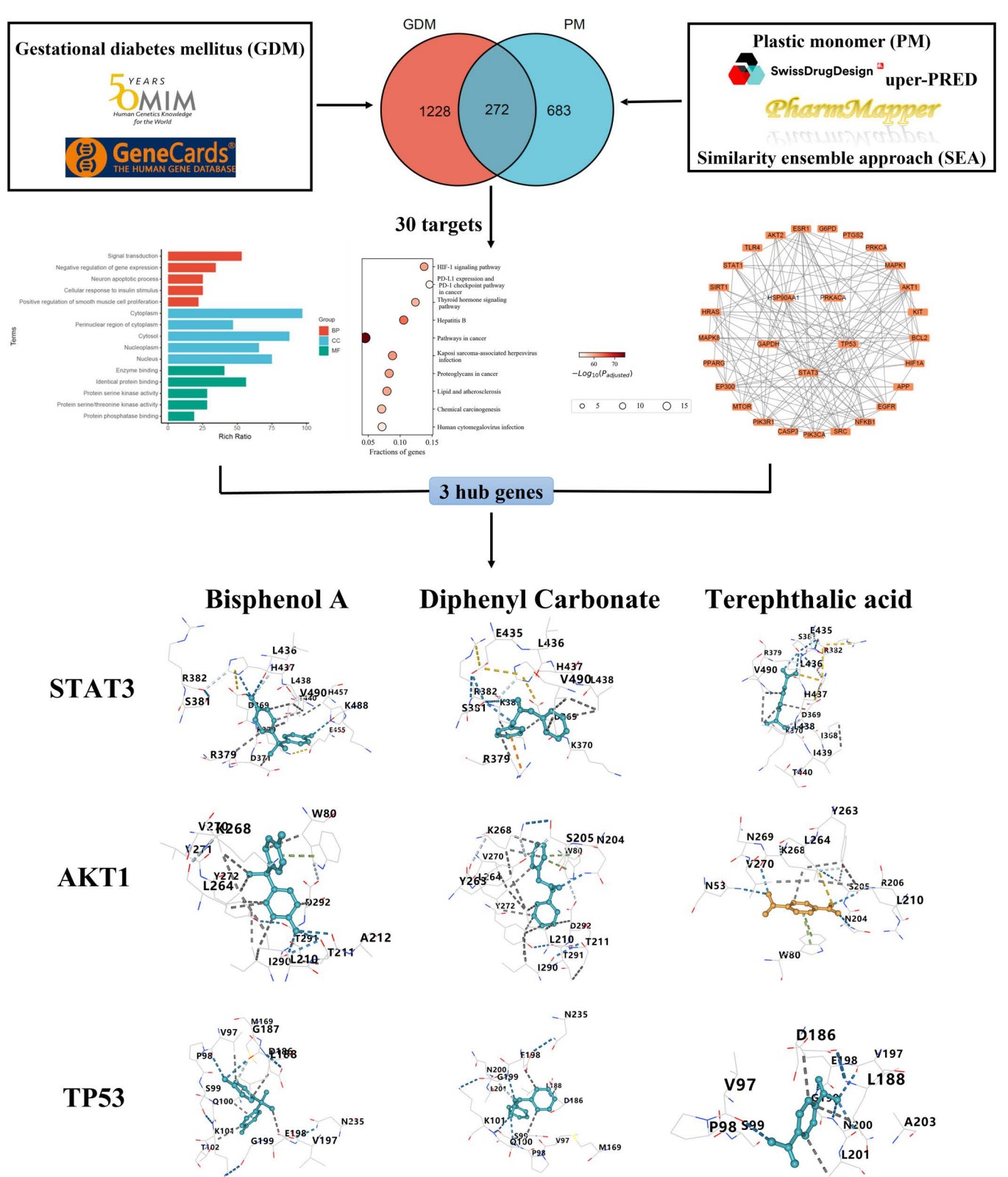

**Fig 1. Flowchart of this work.**

**Table 1. Detailed information of PM.**

| Molecular Name | CAS ID | Formulas | SMILES structures |
|---|---|---|---|
| Acrylonitrile | 107-13-1 | $C_3H_3N$ | C＝CC#N |
| Adipic acid | 124-04-9 | $C_6H_{10}O_4$ | C(CCC(=O)O)CC(=O)O |
| Bisphenol A | 80-05-7 | $C_{15}H_{16}O_2$ | CC(C)(C1＝CC＝C(C＝C1)O)C2＝CC＝C(C＝C2)O |
| Butadiene | 106-99-0 | $C_4H_6$ | C＝CC＝C |
| Caprolactam | 105-60-2 | $C_6H_{11}NO$ | C1CCC(=O)NCC1 |
| Diphenyl Carbonate | 102-09-0 | $C_{13}H_{10}O_3$ | C1＝CC＝C(C＝C1)OC(=O)OC2＝CC＝CC＝C2 |
| Ethylene glyco | 107-21-1 | $C_2H_6O_2$ | C(CO)O |
| Hexamethylenediamine | 124-09-4 | $C_6H_{16}N_2$ | C(CCCN)CCN |
| Methyl Methacrylate | 80-62-6 | $C_5H_8O_2$ | CC(=C)C(=O)OC |
| Styrene | 100-42-5 | $C_8H_8$ | C＝CC1＝CC＝CC＝C1 |
| Terephthalic acid | 100-21-0 | $C_8H_6O_4$ | C1＝CC(=CC＝C1C(=O)O)C(=O)O |

## 2.2 Acquisition of GDM related targets

GDM associated targets were extracted from the GeneCards (https://www.genecards.org/) and OMIM (https://www.omim.org/) databases using the keyword GDM, restricted to Homo sapiens. We selected GDM targets with a relevance score ≥ 10 from the GeneCards database. Duplicates between the two databases were merged and deduplicated. The intersection of PM targets and GDM-related targets was identified as potential key mediators of PM toxicity in GDM pathogenesis.

## 2.3 Construction of protein-protein interaction networks

Common targets shared between PM and GDM were analyzed using the STRING database (https://cn.string-db.org/) to construct a protein-protein interaction (PPI) network with a high-confidence interaction threshold (confidence score ≥0.9) [29]. The network was visualized and analyzed using Cytoscape_v3.10.3 software, with key hub genes identified via the Maximal Clique Centrality, Maximum Neighborhood Component, Degree, Closeness, Betweenness and Stress algorithm.

## 2.4 GO and KEGG enrichment analysis

Gene Ontology (GO) and Kyoto Encyclopedia of Genes and Genomes (KEGG) pathway enrichment analyses of the intersection targets were performed using the DAVID database (https://david.ncifcrf.gov/) to elucidate biological processes, molecular functions, cellular components, and signaling pathways associated with PM induced GDM [30].

## 2.5 Molecular docking and visualization

Core target proteins were retrieved from UniProt, filtered for reviewed and Homo sapiens entries, and their 3D structures were obtained from the RCSB PDB database (https://www.rcsb.org/). Crystal structures were prioritized based on sequence completeness, ligand diversity, and resolution (<3.0 Å). Molecular docking between PM and core targets was performed via CB-Dock2 (https://cadd.labshare.cn/cb-dock2/php/index.php), an AutoDock Vina-based platform enabling automated binding site prediction and affinity scoring [31]. Binding energy served as a key indicator to evaluate the interaction between PM and the core targets: a binding energy below 0 kcal/mol indicates spontaneous binding, while a value lower than −5 kcal/mol suggests stable binding [32,33]. Finally, the conformation with the lowest binding energy was selected for visualizing intermolecular interactions.

## 3. Result

### 3.1 Identification of targets for PM-induced GDM

Through systematic data integration and curation from multiple databases, we identified 955 unique protein targets associated with the 11 PM chemical (Fig 2A). The target profiles exhibited substantial variation among components: acrylonitrile (169 targets), adipic acid (472 targets), bisphenol A (333 targets), butadiene (129 targets), caprolactam (262 targets), diphenyl carbonate (344 targets), ethylene glycol (91 targets), hexamethylenediamine (276 targets), methyl methacrylate (114 targets), styrene (124 targets), and terephthalic acid (372 targets). This comprehensive target mapping reveals the diverse molecular interactions underlying PM toxicity. These candidate target proteins may mediate the toxicological effects of PM exposure, potentially contributing to GDM pathogenesis. Functional characterization and mechanistic validation of these targets could provide critical insights into the molecular basis of PM-induced metabolic dysregulation, thereby facilitating the development of targeted intervention strategies.

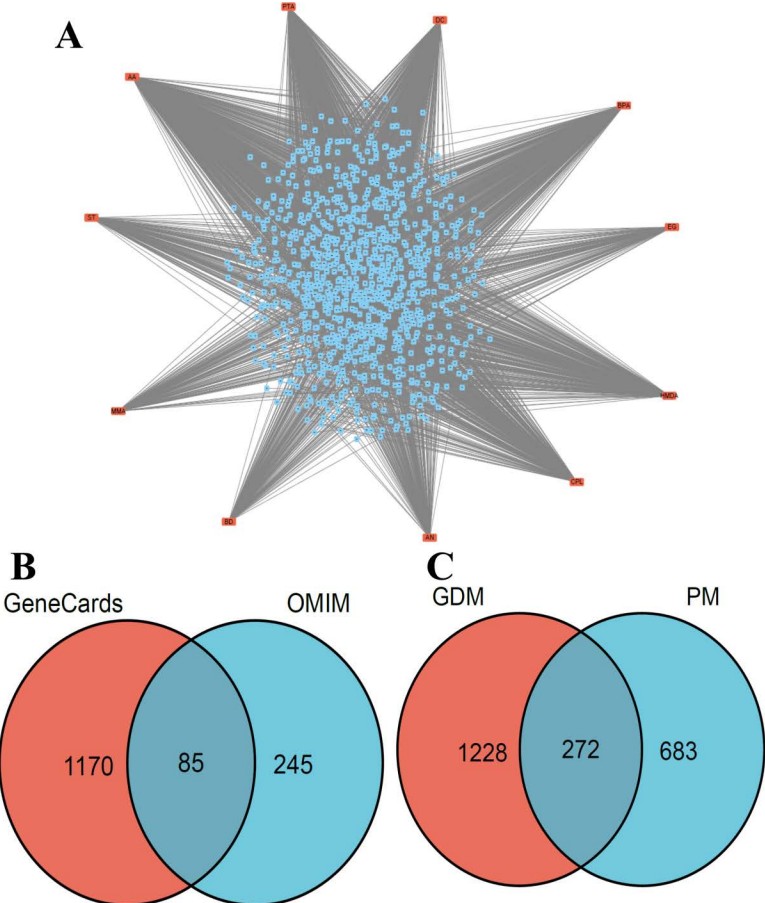

**Fig 2. Identification of common targets between PM components and GDM. (A)** Association network of 11 PM components and their predicted target genes. PM (orange nodes) and their targets (blue nodes) are shown. Abbreviations: AN, acrylonitrile; AA, adipic acid; BPA, bisphenol A; BD, butadiene; CPL, caprolactam; DC, diphenyl carbonate; EG, ethylene glycol; HMDA, hexamethylenediamine; MMA, methyl methacrylate; ST, styrene; PTA, terephthalic acid. **(B)** Venn diagram of GDM targets. **(C)** Overlap between PM-related and GDM gene sets. The intersection identifies 272 shared targets.

Our integrated database analysis identified 1,500 GDM-related targets from GeneCards and OMIM databases (Fig 2B). Comparative analysis revealed 272 overlapping targets between GDM and PM exposure using venn diagram visualization (Fig 2C), representing potential molecular mediators of PM-induced GDM pathogenesis. These results provide mechanistic evidence supporting PM as an environmental risk factor for GDM progression, and prioritized candidate genes for subsequent functional validation studies.

### 3.2 Network-based identification of key molecular targets and core regulatory genes

For mechanistic exploration of these candidate targets, we established a PPI network through the STRING database (confidence threshold: 0.900), yielding 226 potential targets associated with both PM exposure and GDM pathogenesis. The network was imported into Cytoscape for further analysis. Hub targets were identified based on three key topological parameters: Betweenness Centrality, Closeness Centrality, and Degree, with selection criteria set at values above their respective network averages. A total of 30 hub targets were identified, reflecting their central and influential positions within the PPI network. The topological organization of these hub targets is visually summarized in Fig 3A, illustrating their likely regulatory roles in the pathological processes linking PM exposure to GDM.

Network topology analysis was performed using the cytoHubba plugin in Cytoscape to identify pivotal molecular targets. Six distinct centrality algorithms (Maximal Clique Centrality, Maximum Neighborhood Component, Degree, Closeness, Betweenness, and Stress) consistently ranked STAT3, TP53, and AKT1 as the top three hub nodes (Fig 3B). The robust convergence of these computational approaches highlights the network centrality and potential pathological significance of these targets in PM-associated GDM. Their prominent topological positions across multiple analysis methods suggest these molecules may represent key therapeutic intervention points for mitigating PM-induced GDM.

### 3.3 Functional enrichment analysis of candidate targets

Further analysis of the 30 core targets was conducted using GO and KEGG enrichment analysis via the DAVID database, with a focus on Homo sapiens. This analysis generated extensive datasets, including 164 KEGG entries and 223 GO entries, comprising 139 biological processes, 18 cellular components, and 66 molecular functions. All significant entries were identified with a false discovery rate cutoff of < 0.05. From the GO enrichment results, the top five most significant terms for biological processes, cellular components, and molecular functions were selected, and a bar plot of GO enrichment analysis was created (Fig 4A-4B). Biological processes categories were primarily associated with signal transduction and cellular function regulation. Cellular components categories were mainly related to cytoplasm, cytosol and nucleoplasm. Molecular functions categories primarily focused on enzyme binding and kinase activity. These observations highlight the contributions of the core targets to key biological processes, cellular distribution patterns, and functional mechanisms, providing essential guidance for further mechanistic studies.

For the KEGG enrichment analysis, the top 10 significantly enriched pathways were visualized (Fig 4C). The results revealed a predominant enrichment of signaling cascades, chemical carcinogenesis-related pathways, and viral infection-associated pathways, suggesting their potential involvement in GDM pathogenesis. Mechanistically, these pathways may contribute to GDM development through insulin resistance, inflammatory activation, and metabolic disturbances. These findings enhance our understanding of the molecular mechanisms linking PM exposure to GDM.

### 3.4 Molecular docking of PM with core target proteins

The cytoHubba plugin was employed to prioritize key hub proteins, with STAT3, TP53, and AKT1 emerging as the top-ranked candidates based on network centrality scores. Molecular docking simulations were then performed to investigate their potential binding interactions with PM. The binding energies derived from these analyses were graphically represented as a heatmap (Fig 5A), facilitating the evaluation of PM-protein binding preferences. All binding energies were less than 0 kcal/mol, indicating spontaneous binding interactions.

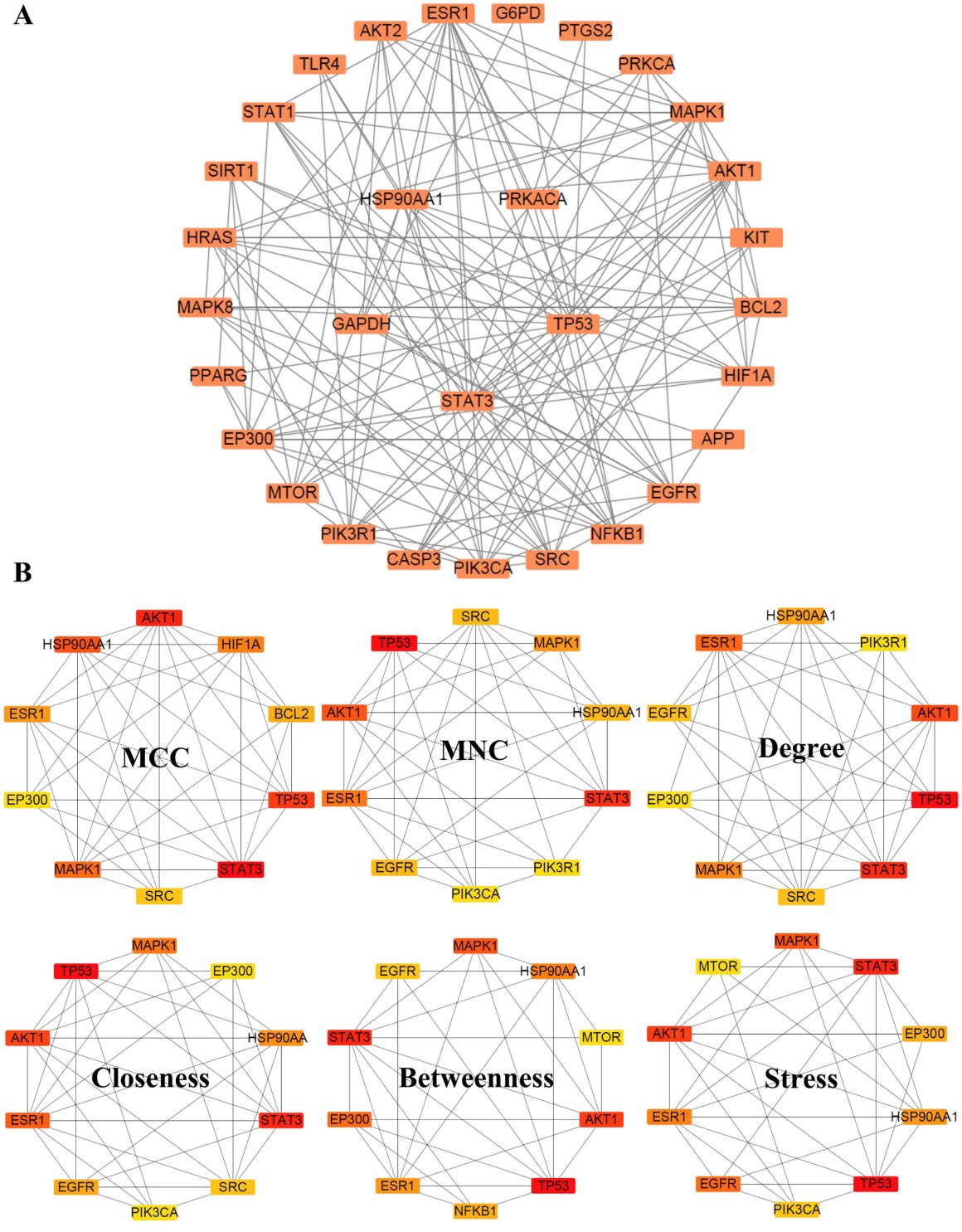

**Fig 3. PPI networks of shared and core targets between PM exposure and GDM. (A)** PPI network of the 30 potential shared targets (30 nodes and 163 edges). Key hubs with high connectivity, such as TP53 and STAT3, are highlighted. **(B)** PPI network of the 10 core targets. Node size and color intensity represent the degree of connectivity, with larger and brighter nodes corresponding to higher degrees.

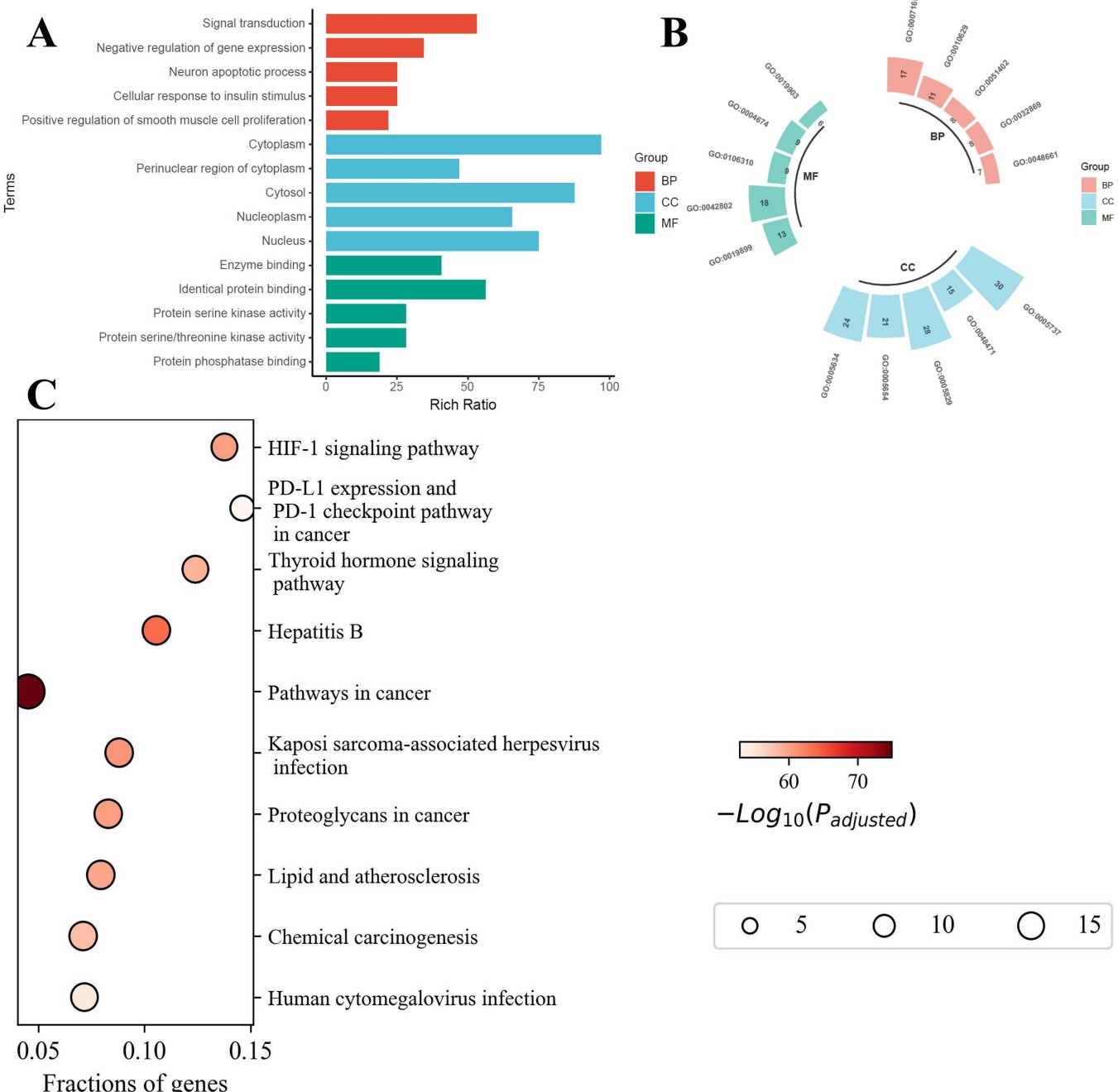

**Fig 4. Functional and pathway enrichment analysis of PM-GDM core targets. (A)** GO enrichment analysis showing significantly enriched terms in biological process (red), cellular component (blue), and molecular function (green). The bar length represents the enrichment fraction for each term. **(B)** Circular bar plot visualizing the number of genes enriched in each GO term. **(C)** KEGG pathway enrichment analysis of the top 10 significantly enriched pathways. Circle size corresponds to the number of enriched genes, and color intensity represents the adjusted p-value, with more intense colors indicating higher statistical significance.

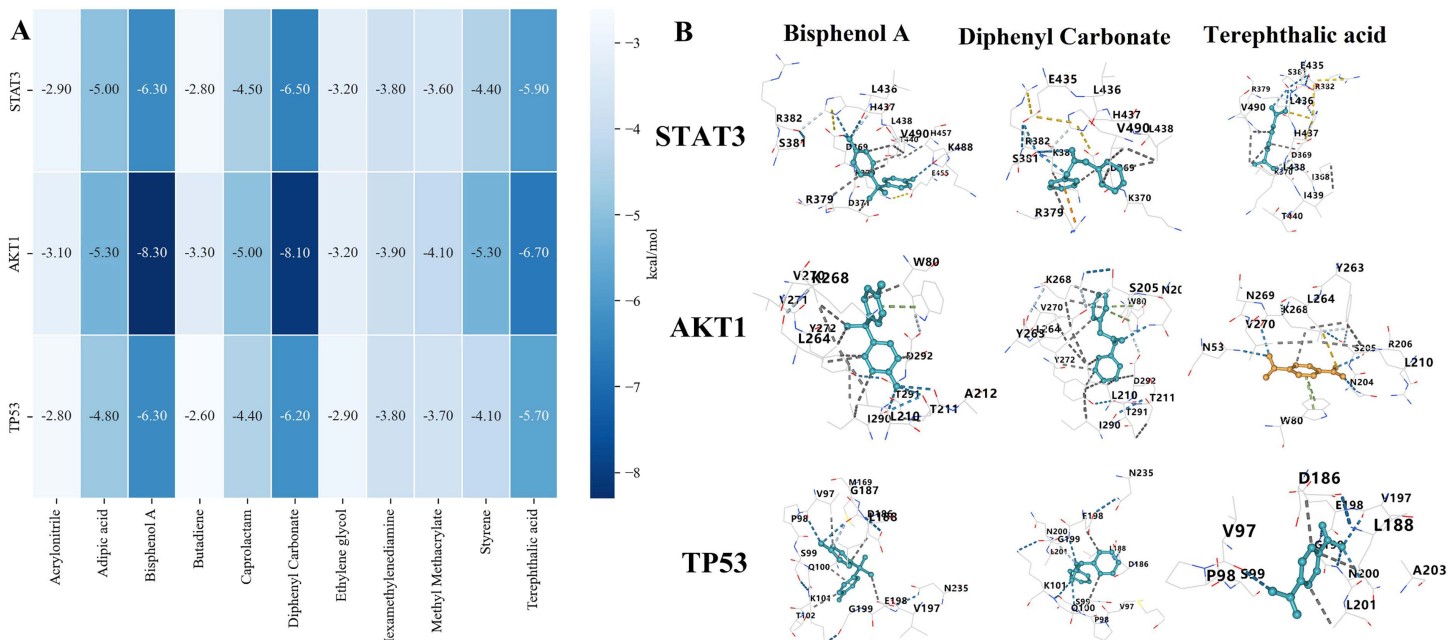

**Fig 5. Molecular docking analysis of PM components with GDM-associated protein targets. (A)** Heatmap of binding energies between PM components and target proteins, with darker colors indicating stronger binding affinity (lower binding energy). **(B)** Detailed visualization of the binding modes of representative PM components (bisphenol A, diphenyl carbonate, and terephthalic acid) within the active sites of core target proteins (STAT3, AKT1, TP53). Molecular interactions with key amino acid residues are highlighted.

Molecular docking analysis demonstrated that bisphenol A, diphenyl carbonate, and terephthalic acid exhibited binding interactions with STAT3, TP53, and AKT1, whereas adipic acid, caprolactam, and styrene selectively bound to AKT1, with adipic acid also interacting with STAT3. The binding affinities for all complexes were favorable (<−5 kcal/mol), indicative of high stability. Further structural analysis revealed that these PM formed a minimum of two hydrogen bonds with key residues of each target protein, suggesting robust molecular interactions (Fig 5B). These findings imply that the high-affinity binding of PM constituents to these core targets may critically contribute to the pathogenesis of GDM following chronic exposure.

## 4. Discussion

The present study systematically elucidated the molecular mechanisms by which PM exposure contributes to GDM through an integrated approach combining network toxicology and molecular docking. The results indicate that PM may interfere with the functions of key proteins such as STAT3, AKT1, and TP53, which in turn perturbs the HIF-1 and thyroid hormone signaling pathways, ultimately leading to GDM. These findings provide novel molecular-level evidence for understanding the association between the environmental pollutant PM and GDM.

At the protein interaction level, molecular docking confirmed that PM can bind to and interfere with the functions of TP53, AKT1, and MAPK1 proteins. TP53, AKT1, and MAPK1 have been reported to play critical roles in the pathogenesis and progression of GDM. The pathophysiological mechanisms of GDM involve impaired insulin secretion and signaling, and its progression is closely associated with multiple interrelated pathological processes, including inflammation, insulin resistance, and oxidative stress [34–36].

The transcription factor STAT3, as a downstream effector of various cytokines (including IL-6, IL-1β, and IL-33), promotes the development of GDM by inhibiting insulin secretion [37]. Through kinase-dependent activation (e.g., JNK,

mTOR, PKCδ), nuclear translocation, and regulation of target gene transcription (such as SOCS family proteins), STAT3 significantly affects insulin sensitivity and promotes IRS1 ubiquitination and degradation, thereby disrupting insulin signal transduction [38,39]. Moreover, STAT3 expression is significantly elevated in the serum and placenta of GDM patients and is strongly associated with GDM occurrence [37]. Aberrant activation of STAT3 represents a core molecular event linking inflammation and insulin resistance in GDM.

AKT1, a key serine/threonine kinase in the PI3K/AKT signaling pathway, is involved in the regulation of glucose metabolism. Its dysfunction is closely associated with metabolic syndromes such as diabetes and dyslipidemia [40]. Dysregulation of this pathway plays a central role in β-cell dysfunction and insulin resistance induced by abnormal lipid metabolism, involving mechanisms related to inflammation and oxidative stress [35]. Studies have shown that genes associated with the PI3K/AKT pathway (e.g., MMPs) are upregulated in placental tissues of GDM patients, and this pathway, together with Jak-STAT and mTOR signaling, participates in diabetes-related pathological processes in GDM [41,42]. Furthermore, multiple studies have demonstrated that modulating this pathway (e.g., via Apelin-13 activation or miR-22/372 inhibition) significantly improves glycolipid metabolism and insulin sensitivity in GDM, suggesting its therapeutic potential [43,44]. Dysfunction of AKT1 directly contributes to metabolic dysregulation in GDM, serving as a key node connecting insulin signaling with glucose and lipid metabolism.

TP53 exacerbates the GDM phenotype by inducing insulin resistance and metabolic dysfunction (e.g., hepatic steatosis) [45,46]. It is closely associated with insulin resistance in adipose tissue and participates in the regulation of insulin sensitivity through the MAPK and NF-κB pathways [47,48]. Stressors such as hyperglycemia, free fatty acids, and inflammation can activate p53, thereby inducing β-cell apoptosis and impairing AKT-mediated cytoprotection, ultimately leading to compromised insulin secretion and glucose homeostasis [35]. Activation of TP53 promotes β-cell damage and insulin resistance in GDM pathogenesis, acting as a critical mediator between stress response and metabolic deterioration.

These core proteins interact within a broader network. For instance, circ-DNMT1 promotes GDM progression by inducing p53 expression and activating the JAK/STAT pathway, while DUSP9 exacerbates insulin resistance and metabolic dysfunction by inhibiting the IRS1/PI3K/AKT pathway [49,50]. Conversely, SIRT1-mediated regulation of STAT3/SOCS3 and the targeting of the STAT3/FAM3A axis by miR-1299 modulate hepatic glucose homeostasis and insulin sensitivity in GDM at different levels [51,52]. Collectively, this evidences a STAT3–AKT1–TP53 core network that integrates multiple GDM pathological mechanisms, including inflammation, oxidative stress, and insulin resistance, offering convergent targets for therapeutic intervention.

Functional enrichment analysis further revealed significant enrichment of the HIF-1 and thyroid hormone signaling pathways in PM-associated GDM. Dysregulation of hypoxia-induced HIF-1 signaling is a known mechanism in the progression of diabetes [53,54]. Studies indicate that the HIF-1 signaling pathway is differentially expressed in the placenta of GDM patients compared to normal controls [55]; HIF-1α expression is closely associated with hyperglycemia, insulin resistance, and inflammation, and it participates in metabolic regulation mediating the development of insulin resistance and diabetes [56].

Thyroid hormone dysregulation, which affects systemic protein, lipid, and glucose metabolism, is epidemiologically linked to an increased risk of diabetes [57]. Thyroid hormones play crucial roles in glucose metabolism and homeostasis, and their dysfunction is considered to contribute to the etiology of GDM [58]. Low thyroid hormone levels in early pregnancy are a risk factor for GDM onset, and abnormal levels of thyroid hormones and related antibodies are frequently observed in GDM patients [59]. Furthermore, elevated levels of free triiodothyronine and total triiodothyronine in early pregnancy have been associated with an increased risk of subsequent GDM development [60]. This suggests that PM may contribute to the pathogenesis of GDM by interfering with the core HIF-1 and thyroid hormone signaling pathways.

Based on existing research, the onset and progression of GDM involve complex interactions among multiple signaling pathways, with the PI3K/AKT/mTOR, PPARγ, and AMPK pathways playing key roles in regulating insulin sensitivity, glucolipid metabolism, and placental function. Multiple studies indicate a close functional relationship between the PI3K/

AKT/mTOR and PPARγ pathways. For instance, pea protein hydrolysate alleviates GDM-related metabolic disorders and placental dysfunction by inhibiting the PI3K/AKT/mTOR/PPARγ signaling pathway [61]. SIRT1 upregulates QKI5 to promote PPARγ expression and activate the PI3K/AKT pathway, thereby ameliorating GDM pathology [62]; Galacto-oligosaccharides significantly improve metabolic parameters in a GDM rat model by modulating the PPARs/PI3K/Akt pathway and gut microbiota [63]. These studies collectively reveal that the PI3K/AKT/mTOR and PPARγ signaling pathways form a functionally synergistic regulatory axis in GDM, representing a critical target for intervening in metabolic disorders.

On the other hand, AMPK, as a cellular energy sensor, also holds a central position in the pathogenesis of GDM. AMPK regulates glucolipid metabolism, protein synthesis, mitochondrial biogenesis, and oxidative stress responses. Its dysfunction is closely associated with maternal hyperglycemia and abnormal fetal development. Studies have shown that AMPK activity is often suppressed in GDM patients, while mTOR signaling is excessively activated, thereby exacerbating metabolic abnormalities [64]. AMPK and AKT often exhibit antagonistic effects under metabolic stress [65], suggesting that they together constitute a finely tuned regulatory network in GDM. Furthermore, AMPK can crosstalk with accumulated p53 protein and HIF-1α signaling under hypoxic conditions [66], although the precise molecular mechanisms require further elucidation. Therefore, impaired AMPK function and its crosstalk with key nodes such as mTOR, AKT, and p53 are central to driving metabolic imbalance in GDM.

Notably, cross-talk extends beyond these pathways. Thyroid hormone receptors interact with nuclear receptors like PPAR, co-regulating cholesterol and glucose metabolism [67]. The competitive relationship between thyroid hormone receptors and PPARα may influence glucose-stimulated insulin secretion [68], hinting at the potential role of thyroid hormone status in GDM development through glucose metabolism. Additionally, research on SGLT-2 inhibitor drugs shows that they exert anti-tumor effects by modulating the AMPK/mTOR and HIF-1α pathways [69], providing a new perspective on the overlap between pathways in metabolic diseases and cancer. This indicates that cross-talk between receptor families and the overlapping pathways across different disease areas offer new insights for a comprehensive understanding of the complex network mechanisms underlying GDM.

In summary, the molecular mechanism of GDM constitutes a multi-pathway, multi-level regulatory network involving key players such as PI3K/AKT/mTOR, PPARγ, AMPK, HIF-1α, and thyroid hormone signaling. The detailed mechanisms of their interactions require further in-depth investigation. We hypothesize that PM contribute to GDM pathophysiology through a cascade of events: initial interference with proteins (e.g., STAT3, AKT1, TP53) and pathways (HIF-1, thyroid hormone) drives inflammatory responses, oxidative stress, and insulin signaling defects, ultimately disrupting the balance of metabolic pathways (PPARγ, AMPK, PI3K/AKT/mTOR) and synergistically leading to insulin resistance and β-cell dysfunction (Fig 6).

Crucially, the core findings of this study are derived from computational biology predictions. These results provide important hypothetical directions for elucidating the potential mechanisms of PM action. However, these predictive outcomes still require subsequent functional validation through in vitro and in vivo experiments to ultimately establish causality. Specifically, the present research has several limitations. First, although the computational predictions are consistent with established biological knowledge, they necessitate further verification via in vitro and in vivo assays. Second, the binding affinities obtained from molecular docking reflect only static binding strengths and do not account for potential allosteric regulatory mechanisms that could alter protein function. Furthermore, this research primarily focused on the toxicity of free monomers and did not comprehensively address the potential composite toxic effects arising from microplastics in real-world exposure scenarios. Microplastics can act as carriers to enhance the biological effects of monomers and may also exert synergistic toxicity with monomers through pathways such as inducing oxidative stress and inflammation. The ubiquitous cocktail effect of environmental pollutants further increases the complexity of the toxic mechanisms. Additionally, constrained by the limitations of the pharmacophore prediction method, this study did not include some common monomers with simple structures and lacking distinctive pharmacophore features (e.g., vinyl chloride $C_2H_3Cl$, ethylene $C_2H_4$), which somewhat impacts the comprehensive assessment of certain widely prevalent monomers.

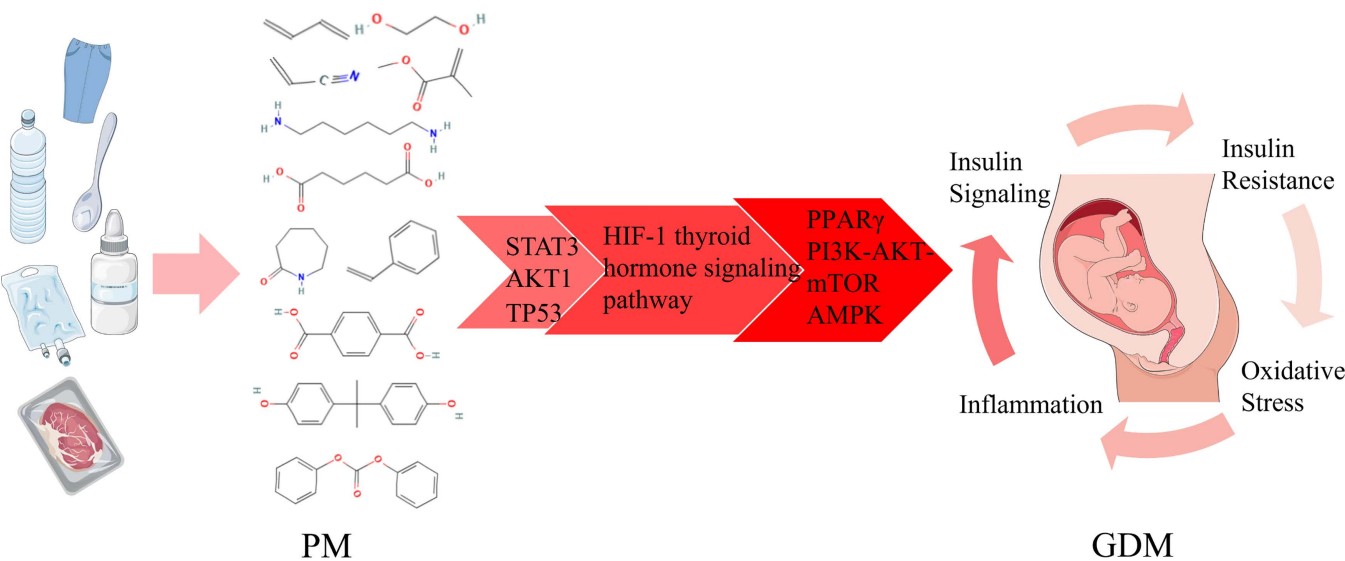

**Fig 6. Proposed mechanistic cascade of PM-induced GDM pathogenesis.**

Future research should focus on constructing a multi-scale assessment framework: employing physiologically based pharmacokinetic models to bridge in vitro and in vivo concentration correlations; developing novel computational methods to enhance the prediction of target interactions for simple-structure monomers; utilizing advanced in vitro model systems to decipher the composite toxicity mechanisms of microplastics and monomers; and integrating exposomics data within population cohorts, simultaneously monitoring PM and microplastic exposure levels. This integrated approach will provide support for achieving a comprehensive risk assessment spanning from molecular mechanisms to population health. Moreover, there is a need to strengthen the epidemiological evidence linking PM exposure to GDM incidence through large-scale longitudinal cohort studies. These should be combined with clinical research to deeply analyze population-specific susceptibility factors (such as gene-environment interactions), thereby providing a scientific basis for formulating precise prevention strategies targeted at high-risk groups.

## Supporting information

**S1 File. Supplementary materials.**
(XLSX)

## Acknowledgments

We express our gratitude to Director Zhao Yun for her leadership and mentorship, as well as to all members of our department for their collaborative efforts and technical contributions throughout this work. This research did not receive any specific grant from funding agencies in the public, commercial, or not-for-profit sectors.

## Author contributions

**Conceptualization:** Tingting Huang.

**Data curation:** Yingying Feng.

**Methodology:** Tingting Huang.

**Project administration:** Yingying Feng.

**Supervision:** Tingting Huang.

**Validation:** Tingting Huang.

**Visualization:** Yingying Feng.

**Writing – original draft:** Yingying Feng.

**Writing – review & editing:** Tingting Huang.

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
