## [Decision Letter · Decision Letter 0]

21 Jul 2025

Dear Dr. Huang,

Thank you for submitting your manuscript to PLOS ONE. After careful consideration, we feel that it has merit but does not fully meet PLOS ONE’s publication criteria as it currently stands. Therefore, we invite you to submit a revised version of the manuscript that addresses the points raised during the review process.

We look forward to receiving your revised manuscript.

Kind regards,

Ruo Wang

Academic Editor

PLOS ONE

Journal Requirements:

4.If the reviewer comments include a recommendation to cite specific previously published works, please review and evaluate these publications to determine whether they are relevant and should be cited. There is no requirement to cite these works unless the editor has indicated otherwise. 

'

Reviewers' comments:

Reviewer's Responses to Questions

**Comments to the Author**

1. Is the manuscript technically sound, and do the data support the conclusions?

Reviewer #1: Partly

2. Has the statistical analysis been performed appropriately and rigorously?

Reviewer #1: N/A

3. Have the authors made all data underlying the findings in their manuscript fully available?

Reviewer #1: Yes

4. Is the manuscript presented in an intelligible fashion and written in standard English?

Reviewer #1: Yes

Reviewer #1: The authors have attempted to make sound conclusive statements in the research paper. However, there are some rivisions that are to be done in order make those conclusive statements. Please go through the reviewr's comments for a more detailed version of improvising suggestions.

**Do you want your identity to be public for this peer review?** For information about this choice, including consent withdrawal, please see our Privacy Policy

Reviewer #1: No

---

## [Author Response · Author response to Decision Letter 1]

1 Sep 2025

Dear Editor and Reviewers,

We would like to express our sincere gratitude to the Editor and Reviewers for their meticulous review and valuable insights on our manuscript (PONE-D-25-16737). Their constructive feedback has been instrumental in enhancing the overall quality and scientific rigor of our paper.

In response, we have revised the manuscript accordingly. Our point-by-point responses are provided below, and the revised manuscript file has been updated to reflect all changes.

Comment 1: The gene targets are treated as independent entities. While the authors have utilized the interaction databases, but they have disregarded gene-gene interaction that can modulate the GDM risk.

Response:

Thank you for your insightful comments and constructive feedback. We fully agree with your observation that gene targets do not function in isolation and that the interaction network among them plays a crucial role in modulating the risk of GDM. This perspective is central to our study, and we have placed significant emphasis on exploring these interactions. Below is a point-by-point response to your concerns:

In our research, we did not treat the gene targets as independent entities. On the contrary, we specifically highlighted that these core targets—STAT3, AKT1, and TP53—collectively mediate the pathogenesis of GDM through a complex protein-protein interaction (PPI) network. Our analytical approach included the following key steps, which directly address your concern regarding gene interactions:

1. Systematic Construction and Analysis of the PPI Network

We used the STRING database to construct a PPI network encompassing all potential targets. The STRING database is a authoritative resource for known and predicted protein interactions. Topological analysis and visualization were performed using Cytoscape, allowing us to identify hub genes within the network. It was through this network-based analysis—rather than evaluating individual genes in isolation—that we concluded STAT3, AKT1, and TP53 represent the most highly connected and functionally central nodes. This indicates that their pathogenic roles are achieved precisely through extensive interactions with other genes and proteins.

2. Detailed Discussion of the Functional Interactions Among Hub Genes

At the molecular level, this study confirmed that plastic monomers (PM) can bind to and interfere with the functions of TP53, AKT1, and MAPK1 proteins. TP53, AKT1, and MAPK1 have been reported to play critical roles in the pathogenesis and progression of GDM. The pathophysiological mechanisms of GDM involve impaired insulin secretion and signaling, and its progression is closely associated with multiple interrelated pathological processes, including inflammation, insulin resistance, and oxidative stress (Kung & Murphy, 2016; Mittal et al., 2025; Saucedo et al., 2023).

The transcription factor STAT3, as a downstream effector of various cytokines (including IL-6, IL-1β, and IL-33), promotes the development of GDM by inhibiting insulin secretion (Li et al., 2023). Through kinase-dependent activation (e.g., JNK, mTOR, PKCδ), nuclear translocation, and regulation of target gene transcription (such as SOCS family proteins), STAT3 significantly affects insulin sensitivity and promotes IRS1 ubiquitination and degradation, thereby disrupting insulin signal transduction (Den Hartogh et al., 2025; Salminen et al., 2021). Moreover, STAT3 expression is significantly elevated in the serum and placenta of GDM patients and is strongly associated with GDM occurrence (Li et al., 2023). Aberrant activation of STAT3 represents a core molecular event linking inflammation and insulin resistance in GDM.

AKT1, a key serine/threonine kinase in the PI3K/AKT signaling pathway, is involved in the regulation of glucose metabolism. Its dysfunction is closely associated with metabolic syndromes such as diabetes and dyslipidemia (Verma et al., 2023). Dysregulation of this pathway plays a central role in β-cell dysfunction and insulin resistance induced by abnormal lipid metabolism, involving mechanisms related to inflammation and oxidative stress (Kung & Murphy, 2016). Studies have shown that genes associated with the PI3K/AKT pathway (e.g., MMPs) are upregulated in placental tissues of GDM patients, and this pathway, together with Jak-STAT and mTOR signaling, participates in diabetes-related pathological processes in GDM (Huang et al., 2024; Zhang et al., 2024). Furthermore, multiple studies have demonstrated that modulating this pathway (e.g., via Apelin-13 activation or miR-22/372 inhibition) significantly improves glycolipid metabolism and insulin sensitivity in GDM, suggesting its therapeutic potential (Li et al., 2022; Zheng et al., 2021). Dysfunction of AKT1 directly contributes to metabolic dysregulation in GDM, serving as a key node connecting insulin signaling with glucose and lipid metabolism.

TP53 exacerbates the GDM phenotype by inducing insulin resistance and metabolic dysfunction (e.g., hepatic steatosis) (Derdak et al., 2011; Punja et al., 2021). It is closely associated with insulin resistance in adipose tissue and participates in the regulation of insulin sensitivity through the MAPK and NF-κB pathways (Geng et al., 2018; Minamino et al., 2009). Stressors such as hyperglycemia, free fatty acids, and inflammation can activate p53, thereby inducing β-cell apoptosis and impairing AKT-mediated cytoprotection, ultimately leading to compromised insulin secretion and glucose homeostasis (Kung & Murphy, 2016). Activation of TP53 promotes β-cell damage and insulin resistance in GDM pathogenesis, acting as a critical mediator between stress response and metabolic deterioration.

Additionally, studies have revealed that circ-DNMT1 promotes GDM progression by inducing p53 expression and activating the JAK/STAT pathway, while DUSP9 exacerbates insulin resistance and metabolic dysfunction by inhibiting the IRS1/PI3K/AKT pathway (Bao et al., 2022; Zhang & Jin, 2025). Conversely, SIRT1-mediated regulation of STAT3/SOCS3 and the targeting of the STAT3/FAM3A axis by miR-1299 modulate hepatic glucose homeostasis and insulin sensitivity in GDM at different levels(Chen et al., 2024; Wen et al., 2025).

These data converge to indicate that STAT3, AKT1, and TP53 do not act in isolation but form a functional synergistic axis. The STAT3–AKT1–TP53 core network integrates multiple pathological mechanisms—such as inflammation, oxidative stress, and insulin resistance—ultimately driving the onset and progression of GDM. This convergence of mechanisms offers potential targets for therapeutic intervention.

3. Pathway Enrichment Analysis Reveals Shared Biological Processes Underlying Interactions

Our KEGG pathway enrichment analysis (e.g., HIF-1 signaling, thyroid hormone signaling pathway) reflects common biological processes involving coordinated actions of multiple genes. These pathways exemplify how functionally related gene sets operate through interactions to carry out specific cellular functions. The significant enrichment of our core targets in these pathways strongly supports the notion that they collectively modulate GDM risk via network interactions.

In summary, the core of our methodological approach is network toxicology, which focuses on how chemical perturbations affect entire biomolecular networks rather than individual targets. Our workflow—from PPI network analysis to hub gene identification and functional interpretation—inherently emphasizes and directly analyzes gene interactions.

We greatly appreciate your comments, which have provided us with an opportunity to clarify the central premise of our study. We have already highlighted the importance of network interactions in the manuscript and have further strengthened these discussions in the revised version. We thank you for your thorough and insightful review

Comment 2: The study focuses on molecular docking results which focuses on active sites only, ignoring the allosteric modulation mechanisms that could alter protein functions. I would suggest the authors to include molecular dynamic simulations study, any wet lab work, or to include and explain the possible allosteric effects of the target protein they are reviewing about. Since there are only three proteins (STAT3, AKT1 and TP53), it would be interesting to add on such contents.

Response:

We thank the reviewer for their insightful and constructive comments. We agree with the valid point raised regarding the limitations of molecular docking, which primarily offers static, binding-site-oriented snapshots and cannot fully capture protein conformational dynamics or allosteric regulatory effects that may be induced by ligand binding. We fully acknowledge that molecular dynamics simulations and experimental validation represent the gold standard for confirming binding mechanisms and functional impacts.

Due to current limitations in experimental facilities and computational resources, we were unable to perform the suggested molecular dynamics simulations or wet-lab experiments within the scope of this study. We sincerely acknowledge this as a current limitation of our work. Should conditions permit in the future, we will prioritize the following studies to thoroughly validate and extend our findings:

1. Computational validation: Performing molecular dynamics simulations to analyze the stability, key conformational changes, and energy profiles of PM complexes with STAT3, AKT1, and TP53, thereby providing deeper insights into potential allosteric mechanisms.

2. Experimental validation: Designing experiments using surface plasmon resonance (SPR) to quantitatively determine binding affinity, along with employing Western blotting and other cellular assays to evaluate the actual effects of PM exposure on phosphorylation levels of key targets and related signaling pathways in cell models.

We believe that, despite current resource constraints, the integrated network pharmacology and molecular docking approach adopted in this study provides a novel and important hypothetical foundation regarding the molecular mechanisms linking PM exposure to GDM, while also identifying clear targets for subsequent validation. We note that in many high-quality studies within the field of network toxicology, similar methodology—primarily relying on molecular docking as a starting point for hypothesis generation—has been adopted during early-stage mechanistic exploration. For instance, Xie et al. in Journal of Translational Medicine used a comparable approach to investigate the molecular mechanisms linking artificial sweeteners to cancer risk (Xie et al., 2025), and Cheng et al. in Ecotoxicology and Environmental Safety applied such methods to explore how triclosan exposure might exacerbate ischemic stroke risk (Cheng et al., 2025). This strategy is well recognized as a common and feasible approach in preliminary mechanistic screening studies, as evidenced by its employment in numerous investigations within the field.

Once more, we express our gratitude for the reviewer’s valuable suggestions, which are greatly instructive for refining our research framework and planning future work.

Comment 3: The construction of the paper is not up to the mark. For example, targets like STAT3 and AKT are labelled as “core targets” without demonstrating their relevance to GDM pathogenesis and its function is general metabolic pathways. It would be better if the authors first explain the functions of the target protein in GDM vs. metabolism.

Response:

We sincerely thank the reviewer for their valuable comments. We fully agree that the identification of STAT3, AKT1, and TP53 as core targets requires thorough biological justification. It is essential to clarify their specific roles in the pathogenesis of GDM, rather than relying solely on computational ranking from network analysis. This will significantly strengthen the conclusiveness and persuasiveness of our findings.

Based on your suggestions, we have supplemented and reorganized the original content of the manuscript. The following explanations, which will be integrated into the Discussion section, provide a solid theoretical foundation—based on prior knowledge—for the determination of these core targets.

At the molecular level, this study confirmed that PM can bind to and interfere with the functions of TP53, AKT1, and MAPK1 proteins. TP53, AKT1, and MAPK1 have been reported to play critical roles in the pathogenesis and progression of GDM. The pathophysiological mechanisms of GDM involve impaired insulin secretion and signaling, and its progression is closely associated with multiple interrelated pathological processes, including inflammation, insulin resistance, and oxidative stress (Kung & Murphy, 2016; Mittal et al., 2025; Saucedo et al., 2023).

The transcription factor STAT3, as a downstream effector of various cytokines (including IL-6, IL-1β, and IL-33), promotes the development of GDM by inhibiting insulin secretion (Li et al., 2023). Through kinase-dependent activation (e.g., JNK, mTOR, PKCδ), nuclear translocation, and regulation of target gene transcription (such as SOCS family proteins), STAT3 significantly affects insulin sensitivity and promotes IRS1 ubiquitination and degradation, thereby disrupting insulin signal transduction (Den Hartogh et al., 2025; Salminen et al., 2021). Moreover, STAT3 expression is significantly elevated in the serum and placenta of GDM patients and is strongly associated with GDM occurrence (Li et al., 2023). Aberrant activation of STAT3 represents a core molecular event linking inflammation and insulin resistance in GDM.

AKT1, a key serine/threonine kinase in the PI3K/AKT signaling pathway, is involved in the regulation of glucose metabolism. Its dysfunction is closely associated with metabolic syndromes such as diabetes and dyslipidemia (Verma et al., 2023). Dysregulation of this pathway plays a central role in β-cell dysfunction and insulin resistance induced by abnormal lipid metabolism, involving mechanisms related to inflammation and oxidative stress (Kung & Murphy, 2016). Studies have shown that genes associated with the PI3K/AKT pathway (e.g., MMPs) are upregulated in placental tissues of GDM patients, and this pathway, together with Jak-STAT and mTOR signaling, participates in diabetes-related pathological processes in GDM (Huang et al., 2024; Zhang et al., 2024). Furthermore, multiple studies have demonstrated that modulating this pathway (e.g., via Apelin-13 activation or miR-22/372 inhibition) significantly improves glycolipid metabolism and insulin sensitivity in GDM, suggesting its therapeutic potential (Li et al., 2022; Zheng et al., 2021). Dysfunction of AKT1 directly contributes to metabolic dysregulation in GDM, serving as a key node connecting insulin signaling with glucose and lipid metabolism.

TP53 exacerbates the GDM phenotype by inducing insulin resistance and metabolic dysfunction (e.g., hepatic steatosis) (Derdak et al., 2011; Punja et al., 2021). It is closely associated with insulin resistance in adipose tissue and participates in the regulation of insulin sensitivity through the MAPK and NF-κB pathways (Geng et al., 2018; Minamino et al., 2009). Stressors such as hyperglycemia, free fatty acids, and inflammation can activate p53, thereby inducing β-cell apoptosis and impairing AKT-mediated cytoprotection, ultimately leading to compromised insulin secretion and glucose homeostasis (Kung & Murphy, 2016). Activation of TP53 promotes β-cell damage and insulin resistance in GDM pathogenesis, acting as a critical mediator between stress response and metabolic deterioration.

Additionally, studies have revealed that circ-DNMT1 promotes GDM progression by inducing p53 expression and activating the JAK/STAT pathway, while DUSP9 exacerbates insulin resistance and metabolic dysfunction by inhibiting the IRS1/PI3K/AKT pathway (Bao et al., 2022; Zhang & Jin, 2025). Conversely, SIRT1-mediated regulation of STAT3/SOCS3 and the targeting of the STAT3/FAM3A axis by miR-1299 modulate hepatic glucose homeostasis and insulin sensitivity in GDM at different levels(Chen et al., 2024; Wen et al., 2025). These findings collectively highlight that the STAT3–AKT1–TP53 core network integrates multiple pathological mechanisms in GDM, including inflammation, oxidative

---

## [Decision Letter · Decision Letter 1]

1 Dec 2025

Dear Dr. Huang,

Thank you for submitting your manuscript to PLOS ONE. After careful consideration, we feel that it has merit but does not fully meet PLOS ONE’s publication criteria as it currently stands. Therefore, we invite you to submit a revised version of the manuscript that addresses the points raised during the review process.

We look forward to receiving your revised manuscript.

Kind regards,

Yanggang Hong

Academic Editor

PLOS ONE

Journal Requirements:

Reviewers' comments:

Reviewer's Responses to Questions

**Comments to the Author**

Reviewer #1: All comments have been addressed

Reviewer #2: (No Response)

Reviewer #3: All comments have been addressed

2. Is the manuscript technically sound, and do the data support the conclusions?

Reviewer #1: Yes

Reviewer #2: Yes

Reviewer #3: Yes

3. Has the statistical analysis been performed appropriately and rigorously?

Reviewer #1: Yes

Reviewer #2: Yes

Reviewer #3: Yes

4. Have the authors made all data underlying the findings in their manuscript fully available?

Reviewer #1: Yes

Reviewer #2: Yes

Reviewer #3: Yes

5. Is the manuscript presented in an intelligible fashion and written in standard English?

Reviewer #1: Yes

Reviewer #2: No

Reviewer #3: Yes

Reviewer #1: (No Response)

Reviewer #2: The manuscript entitled “Exploring the mechanism of plastic monomers on gestational diabetes mellitus based on network toxicology and molecular docking” has many mistakes, authors need to rectify many portions.

• Can sentence transitions be made smoother to avoid abrupt topic shifts between plastics, toxicity, and human health?

• Should the paragraph beginning with “Workers in plastic production…” be reorganized to improve clarity and coherence?

• Can long sentences be split for readability, especially in lines 46–54?

• Should redundant phrases like “significant human exposure” and “widespread human exposure” be revised to avoid repetition?

• Should “leachable plastic compounds” be briefly defined for clarity?

• Is the claim “Plastics are a primary source of human exposure to chemicals” (line 54–55) overstated and in need of justification or clarification?

• Does the link between PM exposure and gestational diabetes mellitus (GDM) need stronger justification in the introduction?

• Are citation numbers missing or out of order in some areas, like where (10) directly jumped to 13, suddenly appearing without context?

• Should older or general citations be replaced with more recent references (last 5–7 years)?

• The author is required to update recent references, which can be seen in PMID: 37065061, 39065802, 35424125, 34297427, 35014595, 36936534.

• Is it necessary to standardise all software names with appropriate citations and versions?

• Should docking parameter details like grid size, scoring function, and validation be added for reproducibility?

• Are all bioinformatics tools properly cited with references?

• Should statistical significance criteria for enrichment analysis (like p-value thresholds or FDR corrections) be mentioned?

• Reinforce statements with references or limit overinterpretation.

• Is the selection of 30 hub genes clearly explained with methodological justification? Clarify criteria and thresholds for selecting top hub genes.

• Has the cutoff for docking binding energies been justified based on literature standards? Explain why ΔG < −5 kcal/mol is considered strong binding.

• Is it clear why 11 PM (plastic monomers) were selected for the study? Explain selection criteria.

• Are abbreviations like AKT1, STAT3, GDM defined at their first appearance? Define all abbreviations when first used.

Good Luck!

Reviewer #3: The authors have invested a great deal of time in revising the paper.

I will still prefer the following.

1. The figures should be placed under the appendices subheadings, with clear references made to them in the report's main body.

2. All figures should have clear legends that enable the reader to follow the diagram.

Apart from the above comments, the paper is good to go.

**Do you want your identity to be public for this peer review?** For information about this choice, including consent withdrawal, please see our Privacy Policy

Reviewer #1: **Yes: ** Darshini S

Reviewer #2: **Yes: ** Shahzaib Ahamad

Reviewer #3: No

---

## [Author Response · Author response to Decision Letter 2]

9 Dec 2025

Response to Reviewer #2

Thank you very much for your valuable comments and careful review of our manuscript. Your feedback is highly pertinent and constructive, and your suggestions will significantly help us improve the quality and clarity of the paper. In response, we have carefully revised and refined the Introduction section of the manuscript based on each of your specific suggestions. Below, we provide a point-by-point response and clarification of the revisions made.

1. Can sentence transitions be made smoother to avoid abrupt topic shifts between plastics, toxicity, and human health?

Response: We thank the reviewers for their thorough evaluation and insightful suggestions for improvement. In response, we have enhanced the logical flow of the manuscript by adding appropriate transitional phrases and linking sentences, thereby smoothing the progression of the narrative from "plastic overview → toxicity mechanisms → health impacts → research gaps." For instance:

1. Following the introduction on the compositional complexity of plastics, we inserted the sentence, "Of particular concern is..." (line 46) to naturally introduce the "releasability" of chemicals as a key prerequisite for toxicity.

2. After enumerating the health risks, we used the phrase, "Such hazards are corroborated by..." (line 58) to transition into the supporting epidemiological evidence.

3. Upon highlighting the imbalance in existing research (which emphasizes additives over monomers), we employed the clause, "Plastics monomers (PM) such as..." (line 69) to directly pivot into a detailed discussion of monomers, thereby avoiding an abrupt shift.

Through the use of more precise connectives and the restatement of key concepts, the revised text now demonstrates improved coherence and a clearer logical progression.

2. Should the paragraph beginning with “Workers in plastic production…” be reorganized to improve clarity and coherence?

Response: The authors thank the reviewers for their specific and actionable suggestions. In response, we have restructured the sentence in question. The original lengthy, list-based format was broken down, and the evidence was explicitly organized into two distinct tiers: "occupational exposure" and "community exposure." This revision enhances both clarity and logical flow.

Revised Example Text (line 58-64):

“Such hazards are corroborated by occupational and community-based epidemiological studies: plastic production workers exhibit significantly higher incidences of cancer and pulmonary diseases compared to the general population; furthermore, residents near plastic manufacturing facilities demonstrate elevated rates of preterm birth, low birth weight, childhood leukemia, asthma, chronic obstructive pulmonary disease, cardiovascular disease, traffic injuries, and mental health issues (1).”

3. Can long sentences be split for readability, especially in lines 46–54?

Response: The authors thank the reviewers for their meticulous assessment and precise suggestions for improvement. In accordance with the recommendation, we have restructured the complex, lengthy sentence originally located in lines 46-54. The revised text now presents the information in separate, logically sequential sentences, thereby enhancing clarity and improving the flow of the argument.

Revised Example Text (line 46-52):

“Of particular concern is the fact that most of these chemicals are not covalently bonded to the polymer matrix. This allows them to be released into environmental compartments through migration, volatilization, or leaching processes at various stages of the plastic lifecycle. Consequently, these substances can transfer into packaged contents (e.g., food), human exposure microenvironments (e.g., indoor air and dust), and natural environments (e.g., water bodies and soil), thereby establishing plastic as a significant source of human exposure to synthetic chemicals (6, 7).”

4. Should redundant phrases like “significant human exposure” and “widespread human exposure” be revised to avoid repetition?

Response: We thank the reviewers for their insightful and constructive comments, which have substantially enhanced the quality of this manuscript. In response, we have thoroughly revised the text to eliminate redundant phrasing where similar concepts were reiterated. For instance, we have varied the terminology by using phrases such as "widespread human exposure" and "ubiquitous human exposure," or, where contextually appropriate, simply referring to "exposure." This revision eliminates unnecessary repetition and enhances the linguistic diversity of the manuscript.

5. Should “leachable plastic compounds” be briefly defined for clarity?

Response: We sincerely thank the reviewers for their valuable comments and suggestions regarding the manuscript. The phrase in question has been removed in the final revised version; therefore, the associated definitional concern is no longer applicable. The core concept that the phrase conveyed—i.e., that chemicals can leach and migrate from plastics—has been integrated into the preceding description of the toxicity mechanisms (see the sentence: "This allows them to be released...") (line 47).

6. Is the claim “Plastics are a primary source of human exposure to chemicals” (line 54–55) overstated and in need of justification or clarification?

Response: We thank the reviewers for their thorough evaluation and valuable suggestions. In response, we have revised the text to enhance its precision. Specifically, in the final version, we have replaced the somewhat absolute and broad assertion "primary source" with the more contextually specific and measured phrase: "...thereby establishing plastic as a significant source of human exposure to synthetic chemicals (6, 7)." The term "significant source" is supported by preceding evidence regarding release pathways and the documented ubiquity of exposure (e.g., the high detection rates of bisphenol A). This revision more accurately reflects the substantial role of plastics in chemical exposure while avoiding the potentially overly definitive implication of the word "primary."

7. Does the link between PM exposure and gestational diabetes mellitus (GDM) need stronger justification in the introduction?

Response: We are grateful to the reviewers for their insightful comments, which helped address critical gaps and strengthen the argument. In response, we have enhanced the logical progression of this section. Prior to introducing gestational diabetes mellitus (GDM), we now first provide an overview, noting that plastic monomers (PM) exhibit established toxicity (including reproductive and carcinogenic effects) and highlighting the knowledge gap regarding their health impact mechanisms in vulnerable populations such as pregnant individuals. Following the discussion of traditional and emerging risk factors for GDM, we have now added a specific bridging sentence that directly links the known toxicological properties of PM to potential mechanisms underlying GDM:

“Notably, bisphenol A and its substitutes have been implicated in GDM risk. ... This evidence suggests that other widely used, structurally diverse PM with potential metabolic-disrupting activity may also play a role in GDM etiology.” (line 94-100)

This sentence leverages the evidence on bisphenol A (a PM) to form a reasoned inference that other widely used PM with potential metabolic-disrupting properties could similarly contribute to GDM pathogenesis. This provides a more direct logical basis and clearer rationale for subsequently focusing our study on the association between PM and GDM.

8. Are citation numbers missing or out of order in some areas, like where (10) directly jumped to 13, suddenly appearing without context?

Response: You are absolutely correct. This discrepancy was a technical oversight caused by the failure to timely update the reference fields when revising the manuscript using a reference management software (EndNote). In the tracked-changes version of the previous revision, the citation sequence was initially coherent; however, an inadvertent error occurred during the final file generation. We have meticulously reviewed and updated the reference field codes throughout the manuscript to ensure that all citation numbers are now accurate and sequential. In the revised version submitted here, this issue has been fully resolved.

We sincerely apologize for any inconvenience this oversight may have caused during your review.

9. Should older or general citations be replaced with more recent references (last 5–7 years)? The author is required to update recent references, which can be seen in PMID: 37065061, 39065802, 35424125, 34297427, 35014595, 36936534.

Response: We sincerely thank the reviewer for the valuable and constructive comments on our manuscript. The reviewer’s suggestion to replace older or more general references with recent publications (within the past 5–7 years) is highly pertinent and justified. We fully agree that citations should be timely and directly relevant, ensuring that the research context and discussion are grounded in the latest developments. In accordance with this principle, we have systematically reviewed and updated the references throughout the manuscript.

The following two main steps were taken:

1. Comprehensive reference update

All citations in the manuscript were carefully evaluated, particularly older or broad references. While preserving citations of classic theories and key original findings where necessary, we have replaced relevant citations wherever possible with high-quality recent publications (2020–2025) that more precisely support the corresponding arguments.

2. Deliberate assessment of the specifically recommended literature

We also paid close attention to the specific references recommended by the reviewer (PMIDs: 37065061, 39065802, etc.) and examined each in detail. These studies are highly specialized, focusing on the structural dynamics, mutational effects, and inhibitor design of Tau‑tubulin kinases (TTBK1/TTBK2) in neurodegenerative diseases such as Alzheimer’s disease.

After careful consideration, we note that the focus of these recommended papers differs significantly from the central theme of our manuscript, which explores the mechanisms of plastic monomers in gestational diabetes mellitus (GDM) from the perspective of network toxicology and molecular docking. In particular: Our study addresses exogenous environmental pollutants (plastic monomers) and their potential impact on a metabolic disorder (GDM). The recommended literature deals with endogenous neurodegenerative disease targets (Tau‑related kinases) and their structure/function.

Given the distinct disease areas (neurodegeneration vs. GDM), core targets (TTBK kinases vs. insulin‑signaling‑related proteins), and scientific questions, directly incorporating these references into our specific context might not be appropriate and could potentially mislead readers.

We fully appreciate, however, that the reviewer’s intention in providing these PMIDs was to encourage the use of cutting‑edge literature. Therefore, in addition to the comprehensive update described above, we specifically searched for and incorporated recent, high‑quality studies that are directly relevant to the methodological application of computational approaches in our research domain. Two new references (now cited as Refs. 32 and 33) have been added, which more accurately support the computational aspects of our work than would citations from the neurodegenerative field.

We believe that this combined approach—systematic updating paired with targeted supplementation—both adheres to the reviewer’s important principle and maintains a high degree of relevance to the specific topic of our study. This revision has substantially enhanced the scientific rigor of the manuscript. We thank the reviewer once again for prompting this important improvement.

10. Is it necessary to standardise all software names with appropriate citations and versions?

Response: We thank the reviewers for their thorough evaluation and constructive suggestions. Providing software version information is crucial for ensuring reproducibility. Accordingly, we have detailed the version numbers of all key software used in the relevant steps of the Methods section. Additionally, the official access links to all bioinformatics online tools employed (such as SEA, SwissTargetPrediction, STRING, and DAVID) are now provided in the main text.

11. Should docking parameter details like grid size, scoring function, and validation be added for reproducibility?

Response: We sincerely thank the reviewers for their valuable comments and suggestions on the manuscript. Docking was performed using the CB-Dock2 platform, which employs the scoring function of AutoDock Vina. The tool automatically predicts binding pockets and generates docking grids, with the grid size typically set to encompass the entire potential active site. For each ligand-receptor pair, the docking calculations were carried out using the default parameters, and the binding free energy (ΔG, in kcal/mol) of the best-ranked binding pose was recorded. The computational workflow and performance of this platform have been validated in its original publication (Ref. #27), and we have followed its standard procedure.

12. Are all bioinformatics tools properly cited with references?

Response: Yes, we have ensured that all bioinformatics tools and databases used in this study are appropriately cited. Specifically, in the Methods section, a key reference for each tool or database is provided immediately after its description, and the full references are listed in the References section. For example, citations are provided for SEA (Ref. 21), Super-PRED (Ref. 22), SwissTargetPrediction (Ref. 23), PharmMapper (Ref. 24), STRING (Ref. 25), DAVID (Ref. 26), and CB-Dock2 (Ref. 27). These citations correspond to the recognized original methodology papers or authoritative resource articles describing each tool or database.

13. Should statistical significance criteria for enrichment analysis (like p-value thresholds or FDR corrections) be mentioned?

Response: The reviewer raises a very important point. We have explicitly supplemented the statistical criteria used for screening significant entries in Section 3.3, "Functional enrichment analysis of candidate targets." The relevant sentence now reads: "All significant entries were identified with a false discovery rate (FDR) cutoff of < 0.05, thereby defining the statistically significant enrichment results."(line 222)

We believe that with the above additions and clarifications, the methodological rigor and reproducibility of the study have been significantly enhanced. We thank the reviewer again for helping improve the quality of our manuscript.

14. Reinforce statements with references or limit overinterpretation.

Response: Thank you for this important suggestion, which has helped us further improve the rigor of our discussion. We fully agree that scientific statements should be based on published evidence and avoid overinterpretation. We have therefore carefully reviewed the manuscript and revised relevant statements to ensure that all key points are sufficiently supported by the literature and to prevent any potential overinterpretation.

Specifically, we have ensured that all important statements in the text—particularly those concerning facts, data, correlations, and inferences—are accompanied by appropriate citations. When extrapolating from specific evidence (e.g., bisphenol A) to broader categories of plastic monomers, we have employed qualifying language such as "suggests" and "may" to indicate that these are reasonable hypotheses based on existing evidence, rather than established facts. This approach strengthens the discussion while maintaining academic caution and preventing overinterpretation.

We believe that, with the above considerations and the current citation support, the arguments in the manuscript are now adequately substantiated while remaining appropriately cautious. We thank you again for prompting this careful review.

15. Is the selection of 30 hub genes clearly explained with methodological justification? Clarify criteria and thresholds for sele

---

## [Editor Report · Decision Letter 2]

15 Dec 2025

Mechanistic study of plastic monomers in gestational diabetes mellitus: a network toxicology and molecular docking approach

PONE-D-25-16737R2

Dear Dr. Huang,

We’re pleased to inform you that your manuscript has been judged scientifically suitable for publication and will be formally accepted for publication once it meets all outstanding technical requirements.

Kind regards,

Yanggang Hong

Academic Editor

PLOS One

---

## [Editor Report · Acceptance letter]

PONE-D-25-16737R2

PLOS One

Dear Dr. Huang,

I'm pleased to inform you that your manuscript has been deemed suitable for publication in PLOS One. Congratulations! Your manuscript is now being handed over to our production team.

Kind regards,

on behalf of

Dr. Yanggang Hong

Academic Editor

PLOS One